



# One-dimensional models of radiation transfer in heterogeneous canopies: A review, re-evaluation, and improved model

Brian N. Bailey[1], María A. Ponce de León[1], and E. Scott Krayenhoff[2]

[1]Department of Plant Sciences, University of California, Davis, Davis, CA USA

[2]School of Environmental Sciences, University of Guelph, Guelph, ON Canada

**Correspondence:** Brian Bailey, bnbailey@ucdavis.edu

**Abstract.**

Despite recent advances in the development of detailed plant radiative transfer models, the large-scale canopy models generally still rely on simplified one-dimensional (1D) radiation models based on assumptions of horizontal homogeneity, including dynamic ecosystem models, crop models, and global circulation models. In an attempt to incorporate the effects of vegetation heterogeneity or "clumping" within these simple models, an empirical clumping factor, commonly denoted by the symbol $\Omega$, is

5 often used to effectively reduce the overall leaf area density/index value that is fed into the model. While the simplicity of this approach makes it attractive, $\Omega$ cannot in general be readily estimated for a particular canopy architecture, and instead requires radiation interception data in order to invert for $\Omega$. Numerous simplified geometric models have been previously proposed, but their inherent assumptions are difficult to evaluate due to the challenge of validating heterogeneous canopy models based on

10 field data because of the high uncertainty in radiative flux measurements and geometric inputs. This work provides a critical review of the origin and theory of models for radiation interception in heterogeneous canopies, and an objective comparison of their performance. Rather than evaluating their performance using field data, where uncertainty in the measurement model inputs and outputs can be comparable to the uncertainty in the model itself, the models were evaluated by comparing against simulated data generated by a three-dimensional leaf-resolving model in which the exact inputs are known. A new model is

15 proposed that generalizes existing theory, is shown to perform very well across a wide range of canopy types and ground cover fractions.



## 1 Introduction

Solar radiation drives plant growth and function, and thus quantification of fluxes of absorbed radiation is a critical component of describing a wide range of plant biophysical processes. Solar radiation provides the energy for plants to carry out photosynthesis, and drives the energy balance and thus the temperature of plant organs (Jones, 2014). As a result, nearly any attempt to quantify plant development in the natural environment involves acquiring information regarding radiation interception. The unobstructed incoming solar radiation flux is relatively easy to measure, however because of the dense and complex orientation of vegetative elements, characterizing leaf-level radiative fluxes is much more challenging (Pearcy, 1989).

Rather than representing the absorption of radiation by individual leaves, radiation transport is most commonly described statistically at the canopy level through the use of models. Using an analogy to absorption of radiation due to a continuous particle-filled medium, classical radiation transfer theory can be readily adapted to quantify radiation transport within a continuous medium of vegetation as pioneered by Monsi and Saeki (1953). Assuming that scattering of radiation is negligible, and that leaf positions follow a uniform random distribution in space, the governing equation for radiation attenuation within a medium of vegetation is given by Beer's law (also called Beer-Lambert law or Beer-Lambert-Boguer law), which predicts an exponential decline in radiation with propagation distance. The importance of this equation in plant ecosystem models cannot be overstated, and is incorporated within nearly every land surface model (e.g., Sellers et al., 1996; Kowalczyk et al., 2006; Clark et al., 2011; Lawrence et al., 2019), crop model (e.g., Jones et al., 2003; Keating et al., 2003; Stöckle et al., 2003; Soltani and Sinclair, 2012), and dynamic vegetation/ecosystem model (e.g., Bonan et al., 2003; Krinner et al., 2005).

Perhaps one of the biggest challenges in the application of Beer's law is that it inherently assumes that vegetation is homogeneous in space, but many, if not most, of the plant systems in which it is applied are not homogeneous. For example, crops, savannas, coniferous forests, and even tropical forests can have significant heterogeneity due to gaps that freely allow for radiation penetration with near-zero probability of interception (Campbell and Norman, 1998; Bohrer et al., 2009). Crop canopies are inherently sparse early in their development, and can remain so in many perennial cropping systems such as orchards and vineyards. A recent study by Ponce de León and Bailey (2019) quantified errors in the prediction of absorbed radiation using Beer's law for a variety of canopy architectures and found that errors could reach 100% in canopies where the between-plant spacing was larger than the canopy height.

An incredibly wide range of approaches of varying complexity have been used to develop radiation transfer models applicable to heterogeneous canopies. The most robust and computationally expensive approach is to explicitly resolve the most important scales of heterogeneity, such as with a leaf-resolving model (e.g., Pearcy and Yang, 1996; Chelle and Andrieu, 1998; Bailey, 2018; Henke and Buck-Sorlin, 2018) or a 3D model that resolves crown-scale (e.g., Wang and Jarvis, 1990; Cescatti, 1997; Stadt and Lieffers, 2000) or sub-crown-scale heterogeneity (e.g., Kimes and Kirchner, 1982; Sinoquet et al., 2001; Gastellu-Etchegorry et al., 2004; Bailey et al., 2014). To further reduce model complexity, a class of geometric models has been developed that explicitly or statistically represents heterogeneity at the scale of plant crowns in order to predict whole-canopy radiation absorption (e.g., Norman and Welles, 1983; Li and Strahler, 1988; Nilson, 1999; Yang et al., 2001; Ni-Meister et al., 2010). A yet more simplified and perhaps the most commonly used approach is to include an empirical





"clumping" factor $\Omega$ in the exponential argument of Beer's law that effectively scales the radiation attenuation coefficient based on the level of vegetation clumping (Nilson, 1971; Chen and Black, 1991; Black et al., 1991). Clumping usually results in an over estimation of radiation absorption when a model based on Beer's law is used (Ponce de León and Bailey, 2019), and thus setting $\Omega < 1$ reduces the effective attenuation coefficient within Beer's law, which corrects for this over estimation.

Despite the wealth of available models for quantifying radiative transfer in heterogeneous canopies, a critical knowledge gap still exists in which it is usually unclear which model is suited for a particular application, and even a general sense of the errors associated with certain model assumptions is often unknown. Models are commonly selected for historical reasons, based on ease of implementation, availability of computational resources versus domain size, or on perceived errors given the particular model assumptions. This uncertainty is driven by the fact that obtaining robust validation data is exceptionally difficult, and

often uncertainty in model inputs is comparable to uncertainty in the model itself. Offsetting errors and coefficient "tuning" can lead to models that perform exceptionally well in a particular case, but may produce unacceptably large errors in another case. These difficulties have led to a number of model inter-comparison exercises in which simulations are performed of synthetic or artificial canopy cases (Pinty et al., 2001, 2004; Widlowski et al., 2007, 2013). This eliminates ambiguity in model inputs to enable a more objective comparison, however, the "exact" solution is still unknown.

This paper presents a critical re-evaluation of the theoretical basis of simplified one-dimensional (1D) models of radiation transfer in heterogeneous canopy based on Beer's law. Due to the difficulties in objectively comparing and evaluating models based on field data, the performance of various models were explored by applying them in virtually-generated canopies where the inputs are exactly known, and comparing against the output of a detailed leaf-resolving model. The goal of the study was to better understand the implications of radiation model assumptions and uncertainty in model inputs in a wide range of canopy

geometries in order to guide model selection in future applications.

## 2 Theory

### 2.1 Modelling radiative transfer in plant canopies

The governing equation for radiation transfer in a participating medium is the radiative transfer equation (RTE; Modest, 2013), which describes the rate of change of radiative intensity along a given direction $s'$

$$\frac{\partial I(\boldsymbol{r};\boldsymbol{s}')}{\partial \boldsymbol{r}} = -\kappa I(\boldsymbol{r};\boldsymbol{s}') - \sigma_s I(\boldsymbol{r};\boldsymbol{s}') + \kappa I_b(\boldsymbol{r}) + \sigma_s \int\limits_{4\pi} I(\boldsymbol{r};\boldsymbol{s}') \left[ \frac{\Phi(\boldsymbol{r};\boldsymbol{s}',\boldsymbol{s})}{4\pi} \right] d\Omega, \qquad (1)$$

where $I(\boldsymbol{r},\boldsymbol{s}')$ is the radiative intensity at position $r$ along the direction of propagation $\boldsymbol{s}'$, $\kappa$ and $\sigma_s$ are the radiation absorption and scattering coefficients of the medium, respectively, $I_b(\boldsymbol{r})$ is the blackbody intensity of emission at position $\boldsymbol{r}$, $\Phi(\boldsymbol{r};\boldsymbol{s}',\boldsymbol{s})$ is the scattering phase function at position $\boldsymbol{r}$ for incident direction $\boldsymbol{s}'$ and scattered direction $\boldsymbol{s}$, and $d\Omega$ is a differential solid angle.





If scattering and emission within the medium are neglected (i.e., $\sigma_s = I_b = 0$), the RTE can be written more simply as

$$\frac{\partial I(\boldsymbol{r};\boldsymbol{s}')}{\partial \boldsymbol{r}} = -\kappa I(\boldsymbol{r};\boldsymbol{s}').\tag{2}$$

This equation can be integrated along $\boldsymbol{s}'$ from a distance of $\boldsymbol{r} = 0$ to $r$ to yield

$$I(r;\boldsymbol{s}') = I(0;\boldsymbol{s}')\exp\left(-\kappa r\right),\tag{3}$$

The attenuation coefficient can be interpreted physically as the cross-sectional area of radiation-absorbing objects projected in

the direction $\boldsymbol{s}'$ per unit volume of the medium. For a medium of leaves, the attenuation coefficient is given by

$$\kappa = G(\boldsymbol{s}')\,a,\tag{4}$$

where $a$ is the one-sided leaf area density (m$^2$ leaf area per m$^3$ canopy), and $G(\boldsymbol{s}')$ is the fraction of total leaf area projected in the direction of $\boldsymbol{s}'$. It is frequently assumed that the attenuation coefficient does not have azimuthal dependence, and therefore the G-function can be written as $G(\theta)$, where $\theta$ is the zenithal angle of the direction of radiation propagation. The probability of

a beam of radiation, inclined at an angle of $\theta$, intersecting a leaf within a homogeneous volume of vegetation after propagating a distance of $r$ can thus be written as

$$P(r) = \left[1 - \exp\left(-G(\theta)\,a\,r\right)\right].\tag{5}$$

For a canopy that extends indefinitely in the horizontal, the propagation distance $r$ can be re-written in terms of the canopy height $h$ as $r = h/\cos\theta$. Substituting this relation for $r$, and noting that $a\,h = L$ (if $a$ is constant), yields the common form of

Beer's law applied to plant canopy systems

$$P = \left[1 - \exp\left(-\frac{G(\theta)\,L}{\cos\theta}\right)\right].\tag{6}$$

## 2.2  Application of Beer's law in heterogeneous canopies

As introduced previously, Eq. 6 is only valid within a homogeneous volume of vegetation, i.e., leaves are uniformly distributed in space. In reality, essentially all canopies have heterogeneity or "clumping" at a number of scales. There is inherent clumping

at the leaf scale, as leaves are discrete surfaces unlike arbitrarily small gas molecules, and thus even at this scale the assumptions




of Beer's law are violated. Above the leaf scale, leaves are grouped around shoots or spurs, creating heterogeneity at a larger scale. Shoots are grouped around main branches or the plant stem, creating yet another scale of heterogeneity. Large-scale clumping typically exists due to gaps between individual trees, or clearings in the canopy.

In a strict sense, each of these scales of heterogeneity violates the assumptions of Beer's law. One obvious means of dealing with this heterogeneity is to use a more complicated model that explicitly resolves the important scales of heterogeneity, such as a "multilayer" model that resolves heterogeneity in the vertical direction (e.g., Meyers and Paw U, 1987; Leuning et al., 1995), a 3D model that resolves plant-scale heterogeneity (e.g., Wang and Jarvis, 1990; Cescatti, 1997; Stadt and Lieffers, 2000), a 3D voxel-based model that resolves sub-plant heterogeneity (e.g., Kimes and Kirchner, 1982; Sinoquet et al., 2001; Gastellu-Etchegorry et al., 2004; Bailey et al., 2014), or a 3D leaf-level model that resolves heterogeneity at the leaf scale (e.g., Pearcy and Yang, 1996; Chelle and Andrieu, 1998; Bailey, 2018; Henke and Buck-Sorlin, 2018). However, each of these models incur a level of computational expense that may be unacceptable for global-scale models or crop models, which require high efficiency that is provided by simpler models based on Beer's law. As a compromise, numerous geometric models have been proposed that calculate radiation interception using analytical geometric solutions along with simplifying assumptions of the basic shape of canopy elements, such as a hedgerow crop (Tsubo and Walker, 2002; Annandale et al., 2004), spherical/ellipsoidal crowns (Norman and Welles, 1983; Li and Strahler, 1988; Ni-Meister et al., 2010), or conical crowns (Kuuluvainen and Pukkala, 1987; Van Gerwen et al., 1987; Li and Strahler, 1988). However, these models do not necessarily generalize to an arbitrary canopy, and in many cases the choice of the average geometric input parameters may be unclear and thus their determination may require an empirical inversion (e.g., Li and Strahler, 1988).

### 2.3 "Ω" canopy clumping factor approach for incorporating vegetative heterogeneity

The issue of incorporating the effects of clumping in Beer's law models gained a heightened level of attention in the early 1990s from investigators looking to use radiation measurements to invert Beer's law for leaf area index (LAI) values (Chen and Black, 1991; Black et al., 1991; Chen and Black, 1993). Canopy non-randomness or clumping causes underestimation of LAI values inferred in this way, and thus there was a pressing need for a theoretical formulation that could remove the effects of clumping within the inversion procedure, which was also mathematically simple enough that it could be easily inverted for LAI. In an early attempt at applying such a correction, Chen and Black (1991) introduced an empirical coefficient within Beer's law which was termed the "clumping index" and denoted by $\Omega$:

$$P = \left[ 1 - \exp\left( -\frac{G(\theta)\Omega L}{\cos\theta} \right) \right]. \tag{7}$$

Chen and Black (1991) references the early work of Nilson (1971), who originally derived this relationship (his Eq. 25, with the "clumping" parameter denoted by $\lambda$). Nilson (1971) derived this relationship for "stands with a clumped dispersion of foliage" using a Markov chain model. The assumption was that the probability of interception at any point also depends to some degree on the probability of interception at a previous location, with the level of dependence described by $\Omega$ (or $\lambda$ using





the notation of Nilson (1971)). This approach, however, does not explicitly allow for large-scale gaps in vegetation such as crown-scale clumping, but rather assumes that there is some quasi-homogeneous medium with regular and repeating variation in vegetation density.

Within a few years, the clumping factor approach was incorporated within radiation transport models (e.g., Chen et al., 1999; Kucharik et al., 1999; Kull and Tulva, 2000), and quickly became ubiquitous in its application within land surface models (Krayenhoff et al., 2014; Han et al., 2015), remote sensing inversion models (e.g., Kuusk and Nilson, 2000; Anderson et al., 2005; Tao et al., 2016), crop models (e.g., Rizzalli et al., 2002; Teh, 2006; Jin et al., 2016), and ecological models (e.g., Walcroft et al., 2005; Ni-Meister et al., 2010). These applications generally necessitate simple and highly efficient models

for radiation interception, and thus Eq. 7 provides a reasonable compromise between model complexity and accuracy. An additional benefit is that it has only one empirical parameter ($\Omega$) to be specified that can incorporate the effects of vegetation heterogeneity.

    Despite the seemingly attractive simplicity of the clumping factor approach, it has many limitations. Foremost of these limitations is that in general the clumping factor $\Omega$ has a strong dependence on every other variable in Eq. 7, namely $\theta$, $L$,

and $G$ (Chen et al., 2008; Ni-Meister et al., 2010). In effect, specifying $\Omega$ requires knowledge of $P$. Thus, if $P$ is already known, there is no need to determine $\Omega$ and apply Eq. 7 in the first place. Empirical modeling of $\Omega$ (e.g., Kucharik et al., 1997; Campbell and Norman, 1998; Kucharik et al., 1999) is effectively a matter of empirically modeling $P$. If the leaf orientation and heterogeneity are both isotropic, then $G$ and $\Omega$ would no longer have $\theta$ dependence. In this case, the product $G\Omega L$ becomes inseparable and the canopy begins to appear homogeneous with augmented $G\Omega L$. The probability of interception can then be

written as

$$P = \left[ 1 - \exp\left( -\frac{\Omega'}{\cos\theta} \right) \right], \tag{8}$$

where $\Omega' = G\Omega L = \text{const}$. Thus, if $P$ is known for any particular $\theta$, $\Omega'$ can be determined.

### 2.4 Geometric modeling of radiative transfer in heterogeneous vegetation

#### 2.4.1 Direct (collimated) radiation component

Beer's law is only explicitly valid in a medium in which the probability of radiation interception is homogeneous over some discrete scale. Thus, in order to apply Beer's law in heterogeneous vegetation, we must segment the canopy into sections over which we can assume that the vegetative elements are homogeneous in space. In a typical canopy, there may be multiple scales over which this is applicable (Fig. 1). At the leaf scale, the probability of interception is approximately homogeneous, and is equal to the leaf absorptivity. For a "clump" of leaves (e.g., a shoot), the probability of interception may be assumed

approximately homogeneous, and is given by Beer's law (Eq. 6). There may be large gaps between individual shoots, which invalidate Eq. 6, but the overall distribution of shoots across a crown may be approximately homogeneous, and thus the





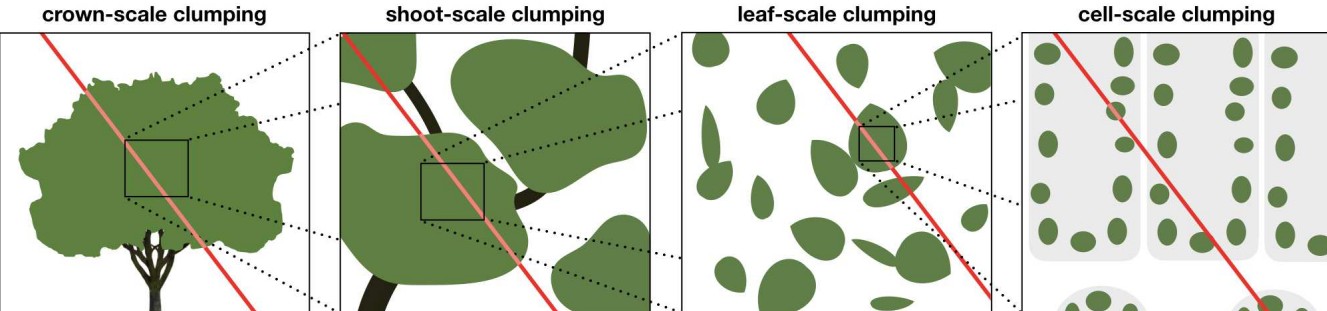

**Figure 1.** Hierarchical scales of spatial aggregation or "clumping" in a plant canopy.

probability that a photon intersects an individual shoot may follow Beer's law but with an augmented attenuation coefficient. Finally, there may be large gaps between crowns that invalidates Eq. 6, but this heterogeneity may be regular and repeating, and thus the probability that a photon intersects an individual crown may be described by Beer's law.

The cumulative probability of photon interception can be viewed as an aggregation of many "clumps" consisting of a homogeneous medium of elements at a smaller scale. For each clump, the probability of a photon intersection is assumed homogeneous in space, and thus the probability of interception can be assumed constant. In this case, the cumulative probability of interception over all clumping levels is

$$P = \prod_{i=1}^{N_c} P_i, \tag{9}$$

where $P$ is the cumulative probability of interception over all scales, $N_c$ is the total number of clumping levels, and $P_i$ is the probability of intersecting the $i^{th}$ clumping level.

     The product of Eq. 9 could be applied in any number of ways depending on the scale at which various probabilities of interception are known. In the example given below, we will follow an approach similar to Nilson (1999) in which the probability of interception within a single plant crown is determined, then repeated $N_c$ times for each crown in the canopy that

the beam of radiation traverses. Accordingly, the canopy is segmented into crown "envelopes" (cf. Nilson, 1992), each of which is conceptualized to a volume encompassing all vegetation within the plant inside which it is assumed that vegetation is homogeneous.

     The probability of a beam of radiation intersecting a single crown is the product of the probability of intersecting the crown envelope and the probability of intersecting a leaf within the envelope. At a solar zenith angle of zero, the probability of

intersecting the crown envelope is given by the ground cover fraction $f_c$, which is the area of the crown envelope shadow at a solar azimuth of zero $S(0)$ divided by the average plan area of ground associated with a single plant. For spherical or cylindrical crowns of radius $R$ and spacing $s$, the ground cover fraction is $f_c = S(0)/s^2 = \pi R^2/s^2$. The probability that a





beam intersects a leaf within a single crown is simply given by Eq. 5, with $r$ being the path length through the crown. Because of the non-linearity of this equation, we cannot simply compute the average $r$ over the crown and substitute it into Eq. 5.

Rather, Eq. 5 should be weighted by the probability that a beam path length through the crown is equal to some value $r$. Thus, the probability $P_\ell$ that a ray passing through a crown envelope intersects a leaf can be written as

$$P_\ell = \int p(r;\theta)\left[1 - \exp\left(-G(\theta)\,a\,r\right)\right] \mathrm{d}r, \tag{10}$$

where $p(r;\theta)$ is the probability that the beam path length through a crown is equal to $r$ for a solar zenith angle of $\theta$. Li and Strahler (1988) showed that for a sphere of radius $R$, $p(r) = r/2R^2$ (no $\theta$ dependence for spheres), with $0 \le r \le 2R$

and thus the limits on the integral in Eq. 10 become 0 to $2R$. For other shapes, analytical expressions for $p(r;\theta)$ become tedious or impossible to derive. However, the integral expressions given by Li and Strahler (1988) for $p(r;\theta)$ can be evaluated numerically. The approach used here was to perform a line-cylinder intersection tests (which are analytical Suffern, 2007), for a large number of lines inclined in the direction of the sun, which allows for population of a probability distribution. A similar approach could be used for ellipsoids (equations given in Norman and Welles, 1983).

In order to use the above expression to calculate the probability of interception for an entire canopy of repeated crowns, we must assume a statistical distribution that describes the probability of intersecting a crown envelope within a canopy. The two distributions that are commonly chosen are the binomial distribution (Nilson, 1971, 1999) or Poisson distribution (Nilson, 1971; Li and Strahler, 1988; Nilson, 1999). If a (positive) binomial distribution is assumed, the interception probability for the entire canopy is

$$P = 1 - (1 - f_c P_\ell)^{N_c}, \tag{11}$$

where $N_c$ is the number of crowns intersected by the radiation beam. Effectively this amounts to saying that the probability of NOT intercepting an individual crown is $1 - f_c P_\ell$, which is compounded $N_c$ times. When the solar zenith is zero, this means that $N_c = 1$ and thus Eq. 11 correctly yields a probability of intersection of $f_c P_\ell$.

If a Poisson distribution is assumed, the interception probability for the entire canopy is

$$P = 1 - \exp\left(-f_c P_\ell N_c\right). \tag{12}$$

Application of this model effectively assumes that crowns act like a homogeneous medium of objects that repeats in all directions, analogous to individual molecules in a gas. When the sun direction is near zenith, this assumption is not necessarily valid as the canopy consists of only a single layer of objects. In this case, the probability of intersection should be $f_c P_\ell$, but Eq. 12 incorrectly gives a value of $1 - \exp\left(-f_c P_\ell\right)$, which is always less than $f_c P_\ell$.





Nilson (1999) estimated $N_c$ as the ratio of the projected crown shadow area $S(\theta)$ to the projected crown shadow area at a solar zenith of zero $S(0)$ (he used the symbol $K$ rather than $N$). For spherical crowns, $S(\theta) = S(0)/\cos\theta = \pi R^2/\cos\theta$. For ellipsoidal crowns with horizontal radius of $R$ and height $H$, $S(\theta) = \pi R^2\sqrt{1 + (H/2R)^2\tan^2\theta}$ (Nilson, 1999). For cylindrical crowns of radius $R$ and height $H$, $S(\theta)$ is well approximated by $S(\theta) = \pi R^2 + 2RH\tan\theta$.

This approach for estimating $N_c$ works well if the crowns are randomly or uniformly distributed in space, and thus $N_c$ is
azimuthally symmetric. If crowns are oriented in rows, the relationship for $N_c$ changes with azimuth. To account for this, we propose the following. Consider the case in which crowns are spaced at a distance of $s_p$ (plant spacing) in the direction of radiation propagation, and spaced at a distance of $s_r$ (row spacing) in the direction normal to the direction of propagation. The azimuthally symmetric model of $N_c = S(\theta)/S(0)$ can be applied, provided that the asymmetry in the actual ground cover fraction is properly accounted for. This is accomplished by 1) calculating the ground cover fraction using the crown spacing
in the direction of radiation propagation, which in this example is $f_c = S(0)/s_p^2$, and 2) multiplying the final interception probability by the ratio of the isotropic crown footprint area ($s_p^2$ in this example) to the actual crown footprint area ($s_p s_r$ in this example).

To generalize this approach, we take the azimuthally symmetric plant spacing $s$ to be $s = s_r\sin^2\varphi + s_p\cos^2\varphi$, where $\varphi$ is the azimuthal angle between the sun direction and the row direction. Equation 11 can then be generalized to

$$P = \frac{s^2}{s_r s_p}\left[1 - \left(1 - \frac{S(0)}{s^2}P_\ell\right)^{N_c}\right]. \tag{13}$$

It is noted that when $s_r = s_p$, Eq. 13 reduces back to Eq. 11. Table 1 provides a summary of inputs and equations needed to implement Eq. 13.

### 2.4.2 Diffuse radiation component

As introduced above, the RTE (Eq. 1) and thus Beer's law (Eq. 6) are only explicitly valid along a single direction of radiation
propagation, and therefore these equations as written can only be applied for collimated radiation (e.g., direct solar radiation). It is common to adapt Beer's law for diffuse radiation conditions by substituting a modified diffuse radiation attenuation coefficient that is usually assumed to be constant for a particular canopy (e.g., DePury and Farquhar, 1997; Wang and Leuning, 1998; Drewry et al., 2010). However, for the reasons above, this approach is not consistent with the assumptions of Beer's law.

A more robust approach for representing incoming diffuse radiation from the sky is to apply Beer's law to any given direction
in the sky and integrate across the upper hemisphere according to

$$P_{\text{diff}} = \frac{1}{\pi}\int\limits_0^{2\pi}\int\limits_0^{\pi/2} f_d(\theta,\phi)P(\theta,\phi)\cos\theta\sin\theta\,d\theta d\phi, \tag{14}$$





**Table 1.** Summary of inputs and equations needed to implement the BINOM model for canopies with spherical or cylindrical crowns.

| Param. | Description | Equation | |
|---|---|---|---|
| | | **Geometric Inputs** | |
| $R$ | Crown radius | | |
| $H$ | Crown height (cylinder) | | |
| $s_p$ | Plant spacing | | |
| $s_r$ | Row spacing | | |
| $L$ | Whole-canopy leaf area index | | |
| $G(\theta)$ | Fraction of leaf area projected in zenithal direction $\theta$ | | |
| | | **Calculated Values (Geometry Specific)** | |
| | | Spherical | Cylindrical |
| $a$ | Within-crown leaf area density | $3Ls_r s_p/(4\pi R^3)$ | $Ls_r s_p/(\pi R^2 H)$ |
| $S(0)$ | Area of a single crown shadow at solar zenith of zero | $\pi R^2$ | $\pi R^2$ |
| $S(\theta)$ | Area of a single crown shadow at solar zenith $\theta$ | $\pi R^2/\cos\theta$ | $\pi R^2 + 2RH\tan\theta$ |
| $p(r;\theta)$ | Probability that within-crown beam path length is equal to length $r$ | $r/(2R^2)$ | Numerical integration |
| | | **General Model Equations** | |
| $N_c$ | Number of crowns traversed by a radiation beam | $S(\theta)/S(0)$ | |
| $s$ | Adjusted effective spacing for canopies with rows oriented at an angle of $\varphi$ relative to the sun | $s_r\sin^2\varphi + s_p\cos^2\varphi$** | |
| $P_\ell$ | Probability of intersecting a leaf within a crown | $\int p(r)\,[1-\exp(-G(\theta)\,a\,r)]\,\mathrm{d}r$ | |
| $P$ | Canopy-level probability of interception | $\dfrac{s^2}{s_r s_p}\left[1-\left(1-\dfrac{S(0)}{s^2}P_\ell\right)^{N_c}\right].$ | |

**If plant spacing is azimuthally symmetric (e.g., random), $s = s_r = s_p$.





**Table 2.** Summary of 1D models of radiation interception considered in this study.

| Model | Equation | Symbol | Assumptions/notes |
|---|---|---|---|
| BINOM | Eq. 13 | —— | Binomial distribution, variable crown path length, random or row arrangement |
| NIL99_B | Eq. A1 | ● | Binomial distribution, constant crown path length, random arrangement (see Nilson, 1999) |
| NIL99_P | Eq. A3 | ● | Poisson distribution, constant crown path length, random arrangement (see Nilson, 1999) |
| NI10_P | Eq. A6 | —— | Poisson distribution, variable crown path length, random arrangement (see Ni-Meister et al., 2010) |
| OM_CON | Eq. 8 | ● | Beer's law with constant clumping factor $\Omega(0)$ |
| OM_VAR | Eq. 7 | ▲ | Beer's law with empirical clumping factor model (see Campbell and Norman, 1998) |
| HOM | Eq. 6 | ⋯⋯ | Standard Beer's law - leaves randomly positioned in space |

where $P_{\mathrm{diff}}$ is the fraction of incoming diffuse radiation intercepted by the canopy, $P(\theta,\phi)$ is the fraction of radiation intercepted by the canopy for radiation originating from the spherical direction of $(\theta,\phi)$ (such as that calculated by Eq. 13), and $f_d(\theta,\phi)$ is a weighting factor to account for anisotropic incoming diffuse radiation. If the canopy is assumed azimuthally
symmetric, this equation reduces to

$$P_{\mathrm{diff}} = 2 \int_0^{\pi/2} f_d(\theta) P(\theta) \cos\theta \sin\theta \, d\theta, \tag{15}$$

where $f_d(\theta)$ is subject to the normalization

$$\int_0^{\pi/2} f_d(\theta) d\theta = \frac{\pi}{2}. \tag{16}$$

Thus, calculation of diffuse radiation interception is simply a weighted average of collimated radiation originating from the
upper hemisphere. In the case of a uniform overcast sky, $f_d = 1$ and thus the amount of collimated radiation originating from some direction $\theta$ is weighted by $\cos\theta \sin\theta$.

## 2.5 Specification of model inputs

For essentially all of the models considered in this work, the model input parameters are 1) the leaf G-function, 2) leaf area index or density, 3) the relative density of crowns, and 4) a mathematical description of the crown envelope. Most commonly,





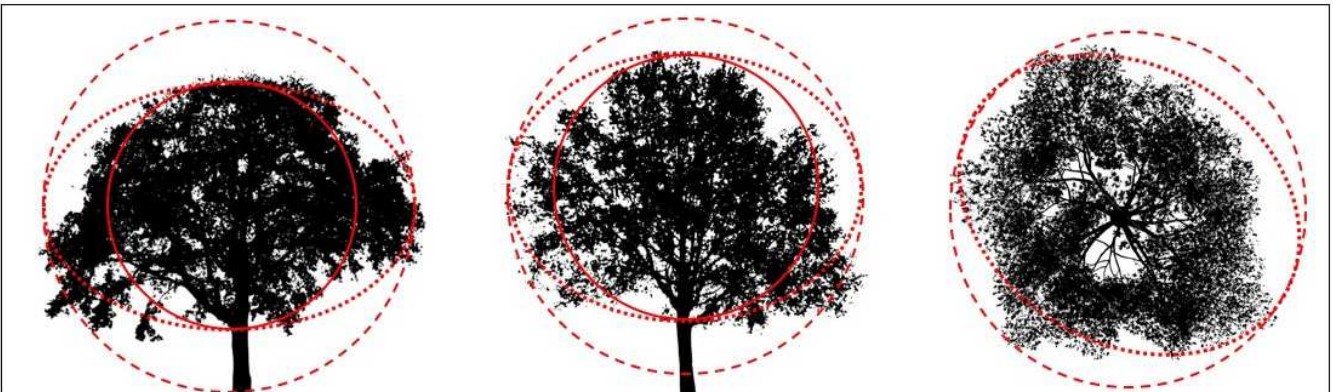

**Figure 2.** Illustration of various crown envelope definitions applied to three different trees. The solid line is a sphere based on the vertical extent of the crown, the dashed line is a sphere based on the horizontal extent of the crown, and the dotted line is an ellipsoid based on the horizontal and vertical extent of the crown.

the crown envelope is assumed to be spherical or ellipsoidal, and thus it is described by its radii. Objectively specifying the crown envelope is generally more of a challenge than it may seem, particularly when recalling that we should define the envelope such that it can be assumed that the vegetation inside the envelope is uniformly distributed in space. Typical crowns have shoots and branches that create an additional scale of clumping, and create irregularly shaped crowns. Figure 2 shows an example of potential spherical and ellipsoidal crown envelopes that could be chosen for a few trees. Clearly, no matter how the

envelope is defined, there is a significant fraction of the envelope that contains open spaces with no leaves, which violates our model assumptions. More complicated envelopes can be derived that better fit the shape of the crown (Nilson, 1999), but these cases require numerical integration in order to calculate the relevant model inputs, namely $S(\theta)$ and $p(r)$.

## 3  Test case set-up

### 3.1  Overview

While it is possible to test the above modelling framework using field data, this approach is severely limited by the lack of systematic variation in canopy architecture as well as experimental errors that can become convoluted with model errors. As such, it can be difficult if not impossible to use field data to rigorously evaluate and diagnose issues within models. In this work, an alternative approach was used to evaluate model performance, which was based on the use of a detailed 3D leaf-resolving model to simulate radiation absorption in virtually-generated canopies. The advantage of this approach is that arbitrary canopy

geometries can be generated in which the exact geometry is known, which provides the necessary inputs for a 1D model. The limitation of course is that results are confined within the assumptions and accuracy inherent in the chosen 3D model. In order to minimize this limitation, the ray-tracing-based model of Bailey (2018) was used as implemented in the Helios 3D modelling framework (Bailey, 2019). Provided that simulated surfaces are isotropic absorbers, Bailey (2018) showed that this





model converges exponentially toward the exact solution as the number of rays is increased. It has also been shown to converge
to the solution given by Beer's law for a truly homogeneous canopy (Ponce de León and Bailey, 2019). Since it was verified
that further increasing the number of rays did not significantly affect results, we considered the 3D model solution to be the
reference or "exact" solution against which the various 1D models could be compared. The canopy-level intercepted radiation
flux was calculated from the 3D model output as

$$
P = \frac{1}{R_\downarrow A_g} \sum_{i=1}^{N_p} R_i A_i,
\tag{17}
$$

where $R_\downarrow$ is the above-canopy solar radiation flux on a horizontal surface, $A_g$ is the horizontal area of the canopy footprint, $R_i$
is the radiative flux incident on the $i^{\text{th}}$ of $N_p$ total vegetative elements, and $A_i$ is the one-sided surface area of the $i^{\text{th}}$ vegetative
element.

A number of test cases were formulated to progressively test different aspects of each of the models given in Table 2. For
each of the test cases below, all surfaces were black, and the ambient diffuse radiation flux was set to zero in order to maintain
consistency with these non-geometric assumptions inherent in Beer's law. The above-canopy solar radiation flux was modelled
using the REST-2 model (Gueymard, 2003). The model was run at an hourly timestep for Julian day 79, and the position on the
earth was the equator. This date and location were chosen because a single diurnal cycle provides a full range of solar zenith
angles ranging from 0 to $\pi/2$. This means that it is not explicitly necessary to evaluate performance under diffuse conditions
because, as was illustrated by Eq. 15, the diffuse radiation flux is simply a weighted average of the flux originating from all
zenithal directions.

Agreement between each of the 1D models and the 3D model were analyzed graphically, and quantitatively using the index
of agreement (Willmott, 1981), which is defined mathematically as

$$
d = 1 - \frac{\sum\limits_{i=1}^{N_t} \left(O_i - M_i\right)^2}{\sum\limits_{i=1}^{N_t} \left(\left|O_i - \overline{O}\right| - \left|M_i - \overline{M}\right|\right)^2},
\tag{18}
$$

where $O_i$ is the $i^{\text{th}}$ of $N_t$ flux values from the 3D model (reference dataset), $M_i$ are flux values predicted by the 1D model, and
an overbar denotes an average over all $N_t$ values.

## 3.2 Case #1: Canopy of solid spheres

In order to isolate the effects of crown-scale clumping, a test case was considered in which the canopy consisted of solid,
opaque spheres of radius $R = 5$ m with varying spacing (Fig. 3a). In the first configuration, spheres were arranged randomly
with an average spacing in the horizontal direction of $s/R = 2, 3, 4,$ and 6. It should be noted that the placement of spheres was







**Figure 3.** Visualization of virtual canopy geometries: (a) Case #1 - solid spheres, (b) Case #2 - solid cylinders, (c) Case #3 - uniform vegetation in spherical crowns, (d) Case #4 - uniform vegetation in cylindrical crowns, (e) Case #5 - tree canopy, and (f) Case #6 - non-tree plant canopy (potato). Wireframe meshes in (e) and (f) show the assumed crown envelope based on a best fit to the exact radiation interception. Surfaces are colored using a pseudocolor mapping of the modeled intercepted radiation flux (W m$^{-2}$).





not truly random, as crowns were not allowed to overlap. The second configuration placed the spheres in a non-random row

orientation in which the plant spacing in the row-parallel direction was $s_p/R = 2$, 3, 4, and 6 and the row spacing was $s_r/R = $

4, 6, 8, and 12. Rows were oriented in either the north-south or east-west directions, and crowns were not allowed to overlap.

The spheres were positioned on the ground surface, and thus the canopy height was $h = 2R$. The 1D models were evaluated

with $L = \infty$ since the spheres were solid. Only the models BINOM, NIL99_B, and NIL99_P were considered for this case

because they separately account for crown and sub-crown intersection.

### 3.3 Case #2: Canopy of solid cylinders

To test generalization to geometries with anisotropic crowns, a case was considered with a canopy of solid, opaque cylinders

of radius $R$ = 5 m and height $H = 2R$ (Fig. 3b). The set-up was essentially the same as Case #1, with random, north-south,

and east-west arrangements of crowns with the same set of spacings.

### 3.4 Case #3: Canopy of uniformly distributed leaves in spherical crowns

In order to test the combined effects of crown-scale clumping and leaf-scale attenuation, a test case was considered in which the

canopy consisted of spherical crowns containing homogeneous and isotropic vegetation elements (Fig. 3c). Spherical crowns

of radius $R$ = 5 m were generated in which rectangular leaves of size $0.1{\times}0.1$ m$^2$ were arranged randomly inside the crown

with a uniform spatial distribution, and were randomly oriented following a spherical distribution ($G = 0.5$). For brevity, only

the random crown arrangement is presented, which had the same average spacing as in Case #1 above. The leaf area density

within each crown was set at $a = 0.5$ m$^{-1}$, and the whole-canopy LAI can be calculated as $L = \dfrac{4\pi R^3 a}{3s^2}$, which gives LAI

values ranging from 0.3 to 2.6.

### 3.5 Case #4: Canopy of uniformly distributed leaves in cylindrical crowns

Similar to Case #3, an additional case was considered consisting of cylindrical crowns of radius $R$ = 5 m and $H = 2R$, filled

with uniformly distributed leaves (Fig. 3d). All other parameters are the same as Case #3. For cylindrical crowns, the whole-

canopy LAI is $L = \dfrac{\pi R^2 H a}{s^2}$, which gives LAI values ranging from 0.44 to 3.9.

### 3.6 Case #5: Canopy of trees

In order to test the models for more realistic canopy architectures, canopies of trees were constructed using the procedural tree

generator of Weber and Penn (1995) (Fig. 3e) as implemented in the Helios 3D modelling framework (Bailey, 2019). The trunk

and branches consisted of a triangular mesh of elements forming cylinders, and leaves were rectangles of size $0.12{\times}0.03$ m$^{-2}$

that were masked into the shape of a leaf using the transparency channel of a PNG image of a leaf (see Bailey, 2019). The tree

crown envelope was approximately spherical in shape, but the spatial distribution of leaves was non-uniform. Leaf angles were

sampled from a uniform distribution, and thus $G = 0.5$. Note that in calculating radiation interception, a distinction was not

made between branches or leaves, but rather total attenuation was used. The tree height was approximately $h$ = 6.5 m, and trees





were randomly arranged with average spacing in the horizontal direction of $s$ = 5, 7.5, 10, 15, and 20 m. The canopy leaf area
index values were $L$ = 3.86, 1.57, 0.97, 0.39, and 0.24. An effective crown radius was estimated to be $R$ = 2.9 m, which was
the value that gave the best predictions of radiation interception at a solar zenith angle of zero. Figure 3e shows a visualization
of the assumed crown envelope based on a best fit of both the BINOM and NIL99_BINOM models to the exact intercepted
flux. The leaf area density was variable in space, and thus an effective density within the crown was substituted into the 1D

equations, which was calculated as $a = \dfrac{3Ls^2}{4\pi R^3}$.

### 3.7  Case #6: Canopy of non-tree plants

A potato plant canopy was generated to test the models in more realistic non-tree canopies (Fig. 3f). Similar to the tree canopies,
the stem consisted of a mesh of triangles, and leaves were texture-masked rectangles. The crown envelope could be considered
roughly cylindrical, with leaves of variable size distributed non-uniformly within the cylinder. The effective dimensions of

the crown envelope were estimated to be $R$ = 0.5 m and $H$ = 0.75 m, which is visualized in Fig. 3f. Plants were arranged
randomly, where the average plant spacing was $s$ = 1.1 m, and for the row-oriented canopies $s_p$ = 0.8 m and $s_r$ = 1.5 m. for all
three configurations the canopy LAI was $L$ = 0.8. The effective crown leaf area density was estimated as $a = \dfrac{Ls_p s_r}{\pi R^2 H}$.

## 4  Results

### 4.1  Case #1: Canopy of solid spheres

Figure 4 gives timeseries of the "exact" intercepted radiation flux compared against the simplified 1D models for each of the
canopies of solid spheres with four densities and three plant arrangements. For random plant arrangement (Fig. 4a-d), the
binomial models performed very well for all ground cover fractions, with the best performance occurring at the highest plant
density of $f_c$ = 0.79 ($d \approx 1.0$). The Poisson model significantly under predicted the intercepted flux for most zenith angles,
with the under prediction being largest at $\theta = 0$. The performance of the Poisson model improved as the canopy became less

dense, such that its performance was near that of the binomial model for the sparsest case of $f_c$ = 0.09.

For the east-west row-oriented configuration (Fig. 4e-h), performance of the BINOM model was nearly the same as in the
randomly oriented case. As would be expected, the performance of the NIL99_B model decreased in the east-west row-oriented
case as it assumes an azimuthally symmetric distribution of crowns (e.g., $d$ decreased from about 1.0 to 0.94 in the sparsest
case). The performance of the Poisson model (NIL99_P) actually increased slightly for the east-west orientation, however

this is likely the result of offsetting errors. As evidenced by the NIL99_B results, an east-west row orientation appeared to
cause an over prediction of the intercepted flux, which offsets some of the under prediction due to the assumption of a Poisson
distribution in the probability of crown intersection.

Overall, the BINOM model performed equal to or better than the NIL99_B and NIL99_P for every canopy configuration.
The lowest $d$ value for the BINOM, NIL99_B, and NIL99_P models, respectively, were 0.98, 0.94, and 0.81. It is noted that for





**Table 3.** Summary of test case results. Agreement between the four models of radiation interception is compared against the "exact" interception using the index of agreement (Eq. 18).

| orient. | $f_c$ | index of agreement, $d$ | | | | | | |
|---|---|---|---|---|---|---|---|---|
| | | \multicolumn model | | | | | | |
| | | BINOM | NIL99_B | NIL99_P | Ni10_P | OM_CON | OM_VAR | HOM |
| | | Case #1: solid spheres | | | | | | |
| R | 0.79 | 1.00 | 1.00 | 0.81 | | | | |
| R | 0.35 | 0.99 | 0.99 | 0.90 | | | | |
| R | 0.20 | 0.99 | 0.99 | 0.94 | | | | |
| R | 0.09 | 1.00 | 1.00 | 0.98 | | | | |
| E-W | 0.39 | 1.00 | 0.96 | 0.93 | | | | |
| E-W | 0.17 | 0.99 | 0.94 | 0.93 | | | | |
| E-W | 0.10 | 0.98 | 0.94 | 0.94 | | | | |
| E-W | 0.04 | 0.99 | 0.97 | 0.98 | | | | |
| N-S | 0.39 | 0.98 | 0.97 | 0.82 | | | | |
| N-S | 0.17 | 0.99 | 0.99 | 0.92 | | | | |
| N-S | 0.10 | 0.99 | 0.98 | 0.94 | | | | |
| N-S | 0.04 | 0.99 | 0.99 | 0.99 | | | | |
| | | Case #2: solid cylinders | | | | | | |
| R | 0.79 | 1.00 | 1.00 | 0.86 | | | | |
| R | 0.35 | 0.99 | 0.99 | 0.90 | | | | |
| R | 0.20 | 0.98 | 0.98 | 0.92 | | | | |
| R | 0.09 | 0.99 | 0.99 | 0.96 | | | | |
| E-W | 0.39 | 0.96 | 0.92 | 0.78 | | | | |
| E-W | 0.17 | 0.96 | 0.94 | 0.87 | | | | |
| E-W | 0.10 | 0.99 | 0.98 | 0.94 | | | | |
| E-W | 0.04 | 0.99 | 0.98 | 0.97 | | | | |
| N-S | 0.39 | 1.00 | 0.89 | 0.92 | | | | |
| N-S | 0.17 | 0.98 | 0.90 | 0.90 | | | | |
| N-S | 0.10 | 0.96 | 0.91 | 0.91 | | | | |
| N-S | 0.04 | 0.96 | 0.94 | 0.94 | | | | |
| | | Case #3: spherical crowns | | | | | | |
| R | 0.79 | 1.00 | 1.00 | 0.88 | 0.85 | 1.00 | 1.00 | 0.95 |
| R | 0.35 | 0.99 | 0.97 | 0.98 | 0.95 | 0.99 | 0.82 | 0.66 |
| R | 0.20 | 1.00 | 0.98 | 0.99 | 0.95 | 0.99 | 0.61 | 0.55 |
| R | 0.09 | 0.99 | 0.92 | 0.97 | 1.00 | 0.99 | 0.35 | 0.40 |
| | | Case #4: cylindrical crowns | | | | | | |
| R | 0.79 | 0.99 | 1.00 | 0.89 | 0.75 | 0.99 | 1.00 | 0.98 |
| R | 0.35 | 0.99 | 0.98 | 0.98 | 0.76 | 0.93 | 0.90 | 0.69 |
| R | 0.20 | 0.99 | 0.97 | 0.99 | 0.70 | 0.87 | 0.70 | 0.57 |
| R | 0.09 | 0.99 | 0.85 | 0.90 | 0.64 | 0.74 | 0.40 | 0.41 |
| | | Case #5: trees | | | | | | |
| R | 0.71 | 1.00 | 0.99 | 0.91 | 0.88 | 1.00 | 0.99 | 0.86 |
| R | 0.32 | 0.99 | 0.98 | 0.96 | 0.92 | 0.99 | 0.79 | 0.61 |
| R | 0.18 | 0.98 | 0.97 | 0.98 | 0.94 | 0.98 | 0.54 | 0.49 |
| R | 0.11 | 0.97 | 0.95 | 0.97 | 0.94 | 0.96 | 0.42 | 0.41 |
| | | Case #6: non-tree canopy | | | | | | |
| R | 0.66 | 0.98 | 0.99 | 0.97 | 0.91 | 0.93 | 0.96 | 0.93 |
| R | 0.24 | 0.99 | 0.99 | 1.00 | 0.96 | 0.91 | 0.96 | 0.80 |
| R | 0.11 | 0.99 | 0.98 | 0.99 | 0.98 | 0.90 | 0.95 | 0.76 |
| R | 0.06 | 0.99 | 0.98 | 0.99 | 0.98 | 0.90 | 0.94 | 0.75 |



**Figure 4.** Flux of intercepted radiation versus solar zenith angle for a half diurnal cycle in a canopy of solid spheres (see Fig. 3a). Varying ground cover fractions $f_c$ were simulated (rows in figure) for three different plant arrangements: randomly spaced (a-d), east-west row orientation (e-h), and north-south row orientation (i-l). The "exact" flux is given by the output of the 3D model ( ——— ), which is compared against the 1D models BINOM ( ——— ), NIL99_B (•), and NIL99_P (•).

the case of a canopy of solid objects with random spacing, the BINOM and NIL99_B models are mathematically equivalent, which is confirmed by the results.





### 4.2 Case #2: Canopy of solid cylinders

Figure 5 gives timeseries of the "exact" intercepted radiation flux compared against the simplified 1D models for each of the canopies of solid cylinders with four different densities and three plant arrangements. Trends in model performance were
similar as in the solid sphere case (Fig. 4), except that overall performance for all models was decreased slightly. The lowest $d$ value for the BINOM, NIL99_B, and NIL99_P models, respectively, were 0.96, 0.89, and 0.78. Again, the BINOM model performed exactly equal to the NIL99_B model, and consistently outperformed the NIL99_P model.

### 4.3 Case #3: Canopy of uniformly distributed leaves in spherical crowns

Figure 6 gives timeseries of the "exact" intercepted radiation flux compared against the simplified 1D models for each of the
canopies of randomly spaced spherical crowns with four different average densities. The BINOM model performed very well for all densities, with $d \geq 0.99$. Performance of the NIL99_B was worse than BINOM for this case, as NIL99_B assumes that the radiation path length is constant across the crown cross-section. This assumption had essentially no impact in the densest planting density, but decreased $d$ from 0.99 to 0.92 in the sparsest case. The Poisson models (NIL99_P and NI10_P) showed similar under prediction as in Case #1 (solid spheres), indicating that most of the error originates from the assumption of a
Poisson distribution. The primary difference between the NIL99_P and NI10_P models is that the NIL99_P model assumes the radiation path length is constant across the crown cross-section, which caused a slight increase in the intercepted flux for NIL99_P (as was the case for the NIL99_B versus BINOM models).

The OM_VAR model had very large errors as the canopy became increasingly sparse. For zenith angles near $90°$, this model had an over prediction as large as the homogeneous model, which decreased toward that of the other models as zenith angle
decreased. It should be noted that the OM_VAR is forced to match the exact flux perfectly at $\theta = 0$, and it only needs to model the flux for $\theta > 0$. The OM_CON model, which also is forced to perfectly match the exact flux at $\theta = 0$, performed as well as the BINOM model ($d \geq 0.99$). This model, which assumes a constant impact of heterogeneity for $\theta > 0$, performed much better than the OM_VAR model. Such a result is to be expected, given that for spherical crowns, the crown envelope as well as leaf orientation are isotropic. Thus, since $\Omega(\theta)$ is dependent on both the impact of heterogeneity and $G(\theta)$, the fact that both $G$
and the heterogeneity are approximately constant means that $\Omega$ is approximately constant.

### 4.4 Case #4: Canopy of uniformly distributed leaves in cylindrical crowns

Figure 7 gives timeseries of the "exact" intercepted radiation flux compared against the simplified 1D models for each of the canopies of randomly spaced cylindrical crowns with four different average densities. The anisotropic crown shape acted to better distinguish between the models than for spherical crown shapes. The BINOM model performed very well for all planting
densities ($d \geq 0.99$). As in the case of the solid cylinder canopy, the assumption of constant crown path length resulted in a slight over prediction of the NIL99_B model, which appeared to increase as the canopy became increasingly sparse. Both of the Poisson models (NIL99_P and NI10_P) significantly under predicted the intercepted flux in the densest canopy case. For




**Figure 5.** Flux of intercepted radiation versus solar zenith angle for a half diurnal cycle in a canopy of solid cylinders (see Fig. 3b). Varying ground cover fractions $f_c$ were simulated (rows in figure) for three different plant arrangements: randomly spaced (a-d), east-west row orientation (e-h), and north-south row orientation (i-l). The "exact" flux is given by the output of the 3D model ( ——— ), which is compared against the 1D models BINOM ( ——— ), NIL99_B (●), and NIL99_P (●).



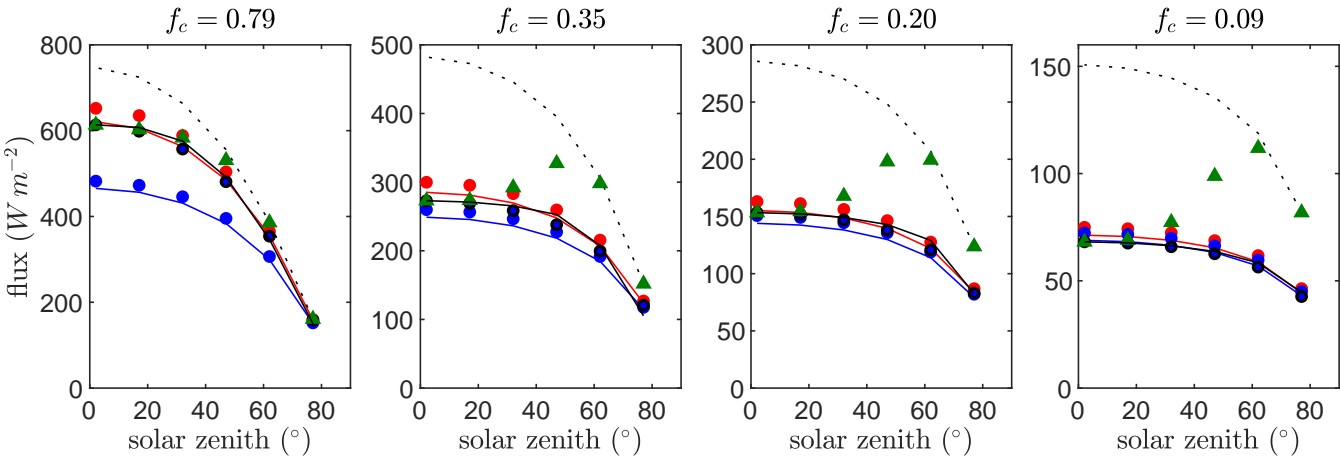

**Figure 6.** Flux of intercepted radiation versus solar zenith angle for a half diurnal cycle in a canopy of spherical crowns filled with uniformly distributed leaves (see Fig. 3c). Varying ground cover fractions $f_c$ were simulated as labeled in each pane. The "exact" flux is given by the output of the 3D model ( ——), which is compared against the 1D models BINOM ( ——), NIL99_B (●), and NIL99_P (●), Ni10_P ( ——), OM_CON (●), OM_VAR (▲), and HOM ( ······).

the sparsest case, the Poisson model NI10_P still significantly under predicted the flux, whereas the NIL99_P over predicted the flux.

The OM_VAR model had a very large over prediction of radiation interception at high zenith angles, as in the spherical crown case. Unlike in the spherical crown case, the OM_CON model did not perform well, particularly as the canopy became increasingly sparse. When crowns are cylindrical, heterogeneity is no longer isotropic especially as the plant spacing becomes large, and as such interception varies irregularly with $\theta$. As a result, $\Omega$ has strong $\theta$ dependence and cannot be assumed constant.

### 4.5    Case #5: Canopy of trees

Figure 8 gives timeseries of the "exact" intercepted radiation flux compared against the simplified 1D models for each of the canopies of randomly spaced trees with four different average densities. Results were quite similar to that of Case #3 (spherical crowns). Model agreement with the reference intercepted flux was reduced slightly overall for each of the models, which is likely due to the fact that the assumptions of spherical crown shape and within-crown homogeneity were not exactly satisfied. However, this reduction in performance was small, and the BINOM model still performed very well. Thus, this test con-
firms generalizability to more realistic tree architectures in which vegetation within the crown envelope is only approximately homogeneous and the crown shape is only approximately spherical.





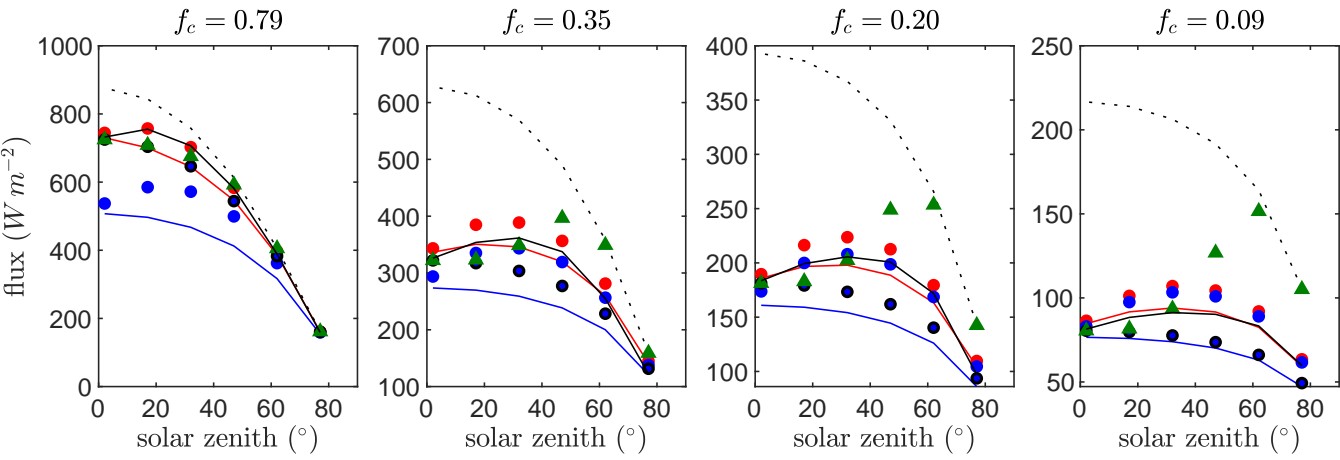

**Figure 7.** Flux of intercepted radiation versus solar zenith angle for a half diurnal cycle in a canopy of cylindrical crowns filled with uniformly distributed leaves (see Fig. 3d). Varying ground cover fractions $f_c$ were simulated as labeled in each pane. The "exact" flux is given by the output of the 3D model ( ——— ), which is compared against the 1D models BINOM ( ——— ), NIL99_B (●), and NIL99_P (●), Ni10_P ( ——— ), OM_CON (●), OM_VAR (▲), and HOM ( ······ ).

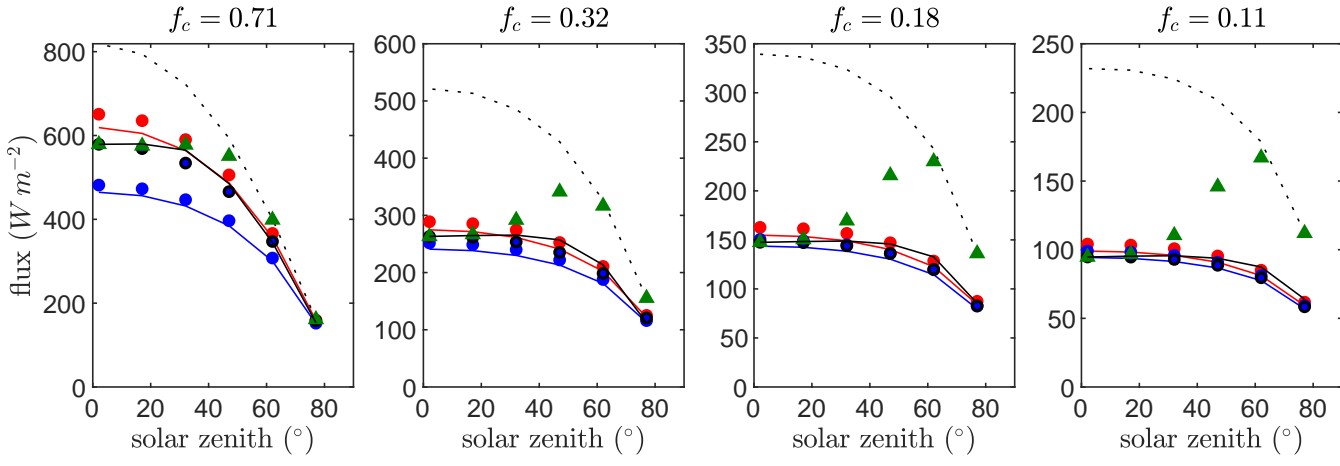

**Figure 8.** Flux of intercepted radiation versus solar zenith angle for a half diurnal cycle in a canopy of trees (see Fig. 3e). Varying ground cover fractions $f_c$ were simulated as labeled in each pane. The "exact" flux is given by the output of the 3D model ( ——— ), which is compared against the 1D models BINOM ( ——— ), NIL99_B (●), and NIL99_P (●), Ni10_P ((eLine), OM_CON (●), OM_VAR (▲), and HOM ( ······ ).

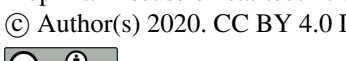



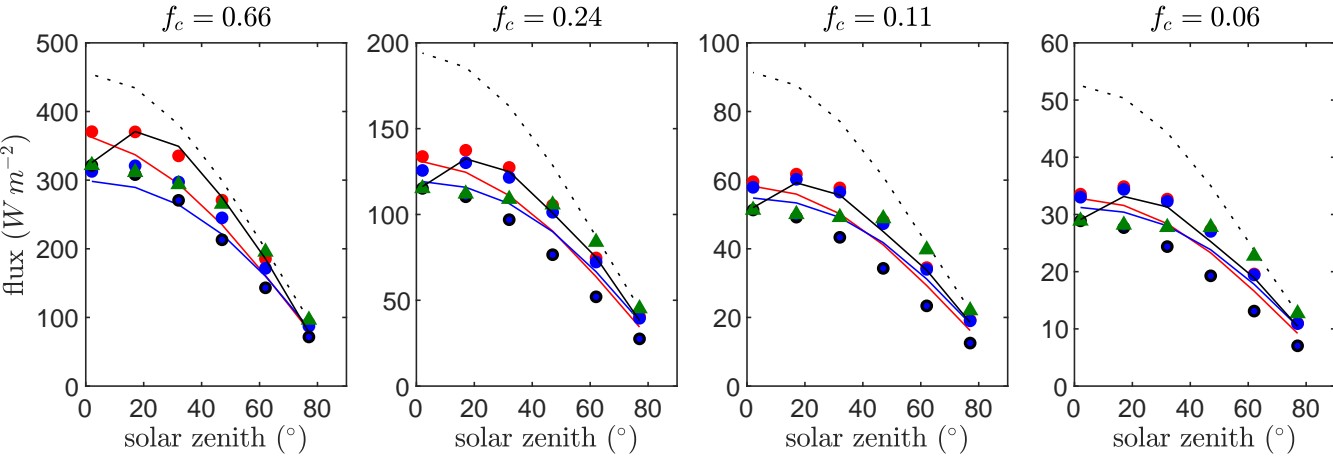

**Figure 9.** Flux of intercepted radiation versus solar zenith angle for a half diurnal cycle in a canopy non-tree plants (potato; see Fig. 3f). Varying ground cover fractions $f_c$ were simulated as labeled in each pane. The "exact" flux is given by the output of the 3D model ( ——), which is compared against the 1D models BINOM ( ——), NIL99_B (●), and NIL99_P (●), Ni10_P ( ——), OM_CON (●), OM_VAR (▲), and HOM ( ······).

## 4.6 Case #6: Non-tree canopy

Figure 9 gives timeseries of the "exact" intercepted radiation flux compared against the simplified 1D models for each of the canopies of randomly spaced non-tree plants (potato) with four different average densities. Results were also similar to the canopy with cylindrical crowns (Case #4), with only slightly reduced overall performance in comparison to Case #4.

## 5 Discussion

The simplest models, based on an $\Omega$ clumping factor that effectively scales the attenuation coefficient, had mixed success depending on the particular case. Assuming a constant $\Omega$ factor (OM_CON model) worked quite well in the case that the heterogeneity (crowns) was roughly isotropic. As mentioned previously, $\Omega$ encapsulates the effects of $G(\theta)$, heterogeneity, and the apparent LAI that results from the heterogeneity to form an inseparable product $G\Omega L$. If this product is isotropic, then it is reasonable to assume a constant $\Omega$. The problem, however, is still that this constant $\Omega$ is not known *a priori* and must be determined from radiation measurements at a particular solar zenith angle (preferably near $\theta = 0$) in order to invert for the appropriate attenuation coefficient.

The OM_VAR model (Campbell and Norman, 1998) attempts to model the effects of anisotropic clumping for $\theta > 0$. When the canopy was fairly dense, the OM_VAR model worked fairly well. However, the OM_CON and HOM models generally performed as well or better than the OM_VAR model for those cases, and thus there is no reason to use a variable $\Omega$ factor. As the canopy became increasingly sparse, the performance of the OM_VAR model declined significantly and over predicted interception as the solar zenith angle increased (note that this model also forces the predicted flux to match the reference





flux exactly at $\theta = 0$). Kucharik et al. (1999) evaluated the framework behind the OM_VAR model in a number of different

canopies and found that it was able to fit the data well, but that the model coefficients were highly species-specific. One issue with their validation approach, which is symptomatic of many field validation studies of heterogeneous canopy radiation models, is that the data was collected in relatively dense canopies where heterogeneity is fairly low overall. The canopies studied in Kucharik et al. (1999) had a ground cover fraction approximately in the range of $f_c = 0.5 - 0.75$, which would place them on the denser end of the canopy cases considered in the present study, which is where the OM_VAR model worked well.

However, it was shown that for these cases a constant $\Omega$ factor model (OM_CON) or even the homogeneous model (HOM) also worked well.

The assumption that the probability of intersecting a crown envelope follows a Poisson distribution did not work well unless the canopy was very sparse. For most canopy densities, the Poisson models significantly under predicted the absorbed flux. It appeared that when the mean free path of radiation propagation (which is related to the crown spacing) was not significantly

smaller than the actual radiation propagation distance through the canopy (which is related to the canopy height and solar zenith), the assumption of a Poisson distribution was poor. The Poisson distribution assumes that the canopy consists of a large number of "layers" of randomly positioned elements. When the solar zenith angle is near vertical, the canopy consists of a single layer of crowns. For solid spherical crowns, the probability of interception should be $f_c$ at $\theta = 0$, but with the Poisson model it is $1 - \exp(-f_c)$, which is always less than $f_c$. As $f_c$ approaches zero, $1 - \exp(-f_c) \approx f_c$, which is demonstrated by

the results of the Poisson model (NI10_P and NIL99_P) evaluation. The NI10_P model has been validated against experimental data by Yang et al. (2010), which showed relatively good model performance. All of the experimental canopies in Yang et al. (2010) were quite dense with high LAI and ground cover fraction. As illustrated previously, specification of the crown radius can be ambiguous for real trees that are irregularly shaped. If the crown radius is specified based on an envelope encapsulating all branches, this effectively inflates $R$ and $f_c$, which would presumably result in an over prediction of the intercepted flux.

However, this over prediction could be offset by applying a Poisson model, which was shown to cause under prediction. It is possible that $R$ was inflated in Yang et al. (2010) (which considered only dense canopy cases), and offsetting errors associated with the Poisson assumption resulted in artificially improved model performance due to offsetting errors. The crown radius in Case #2 is exactly known, so specification of $R$ in that case is not a potential source of model error. In another study, the NI10_P model was compared against other models for a set of virtual canopies where the exact geometry was known (but the

exact radiation interception was not), which suggested that the NI10_P model tended to predict higher canopy transmission (lower interception) than the other models (Widlowski et al., 2013), which is also consistent with the results of the present study.

The assumption that crown intersection followed a binomial distribution appeared to hold for all canopy cases considered in this work. The binomial model predicts the correct interception at $\theta = 0$, which corresponds to a single crown layer ($N_c = 1$)

and $P = 1 - f_c$, rather than $P = 1 - \exp(-f_c)$ in the case of the Poisson model. The BINOM model outperformed all other models for every test considered. The primary difference between the formulation of the BINOM and NIL99_B is that 1) the NIL99_B model assumes that the path length of radiation through an individual crown is constant whereas the BINOM model accounts for variable path length, and 2) the NIL99_B model assumes that crowns are randomly positioned in space and thus





crown intersection is azimuthally symmetric, whereas the BINOM model accounts for asymmetry. The assumption of constant
path lengths created modest errors, as evidenced by the results of Cases #3 and #4. The assumption of random positioning of
crowns in a row-oriented canopy had the potential to create very large errors, as evidenced by the results of Cases #1 and #2.
These errors are caused by the fact that the effective total path length through vegetation in a row-oriented canopy can change
significantly with azimuth.

The scope of the results of this study are clearly limited to cases of no scattering, and no diffuse radiation. These impacts
were excluded from the study to focus on cases where, aside from heterogeneity in the geometry, the assumptions of Beer's
law should be exactly satisfied. Although Beer's law is only valid along a single direction of radiation propagation, and its
derivation requires the removal of scattering terms in the RTE, variations have been derived that approximate the effects of
scattering and diffuse radiation within a 1D model (e.g., Lemeur and Blad, 1974; Goudriaan and Van Laar, 1994).

It appears likely that many crop models, global ecosystem models, and land surface models over estimate radiation intercep-
tion by applying the homogeneous Beer's law in heterogeneous environments, which is sure to have important consequences
for large-scale flux estimates. Incorporation of the results of this work within these models is straightforward, and requires
specification of either the ground cover fraction $f_c$ or the planting density and effective crown envelope. Algorithms are read-
ily available for separation of the ground surface and vegetation within aerial images in order to calculate the ground cover
fraction (e.g., Gougeon, 1995; Luscier et al., 2006; Laliberte et al., 2007). The results of Cases #5 and #6 suggested that rough
estimations of the crown envelope dimensions based on visual inspection could yield reasonable results.

## 6   Conclusion

Simplified models of radiation interception in heterogeneous canopies can be readily derived by separating the canopy into
hierarchical scales of "clumping" over which the probability of interception can be assumed homogeneous in space over
some discrete volume. The results of this work demonstrated that very good predictions of whole-canopy interception can be
achieved using simple geometric models that consider only crown-scale and leaf-scale clumping (in the absence of scattering).
The probability of intersecting a plant crown was well represented by a (positive) binomial distribution. This model calculates
the probability of not intersecting a leaf within a single crown, and compounds this probability $N_c$ times, where $N_c$ is the
number of crowns a given beam of radiation traverses on its path from the top to the bottom of the canopy. The Poisson models
for crown intersection did not perform well unless the canopy was fairly dense, but in this case the effects of heterogeneity are
less important and the homogeneous Beer's law also performs well. The results of the model evaluation exercise confirms that
the binomial model given in Eq. 13 (BINOM) is the preferable model in all cases considered herein. Inputs to the model can be
specified based on measurements of plant geometry, or through inversion if radiation interception measurements are available
for a particular solar zenith angle.





*Code and data availability.* 3D model simulations were performed with Helios version 1.0.14, which can be downloaded from
https://www.github.com/PlantSimulationLab/Helios. Helios project files and output files can be downloaded from
https://doi.org/10.25338/B85C97.

*Author contributions.* BNB conceived the idea for the paper, performed simulations and analysis, and wrote the initial manuscript.
ESK and MAP contributed to theoretical development and design of the study through discussions and editing of the manuscript.

*Competing interests.* The authors declare that they have no conflict of interest.

*Acknowledgments.* Financial support of BNB was provided by the USDA National Institute of Food and Agriculture Hatch
project 1013396, and U.S. National Science Foundation grant #1664175. ESK acknowledges funding from the Natural Sciences
and Engineering Research Council of Canada.

## Appendix A: Descriptions of previously proposed models

For completeness, model equations taken from the literature are provided below as they were implemented and using the
notation adopted in this paper.

### A1  Model of Nilson (1999): NIL99_B and NIL99_P

Assuming that the probability of intersecting a crown envelope follows a (positive) binomial distribution, Nilson (1999) gives
the probability of intersection for a canopy to be

$$P = 1 - \left(1 - (1 - P_1)S(0)/s^2\right)^{N_c},$$ (A1)

where $N = S(\theta)/S(0)$, $S(\theta)$ is the area of the crown envelope shadow, $S(0)$ is the area of the crown envelope shadow at a
solar zenith angle of zero, $s$ is the mean plant spacing, and $P_1$ is the probability that a beam of radiation does not intersect a
leaf within a single crown and is given by

$$P_1 = \exp\left(-\frac{G(\theta)\,L\,s^2}{S(\theta)\cos\theta}\right).$$ (A2)





If the probability of intersecting a crown envelope is instead assumed to follow a Poisson distribution, Nilson (1999) gives
the probability of intersection for a canopy to be

$$P = 1 - \exp\left(-\frac{S(0)}{s^2}(1 - P_1)N_c\right) = 1 - \exp\left(-\frac{S(\theta)}{s^2}(1 - P_1)\right),$$  (A3)

There are two notable differences between Eq. 13 and the model of Nilson (1999). The first was mentioned above, which is that the expression for $N_c$ in Nilson (1999) assumes a random or uniform spatial distribution of crowns, whereas Eq. 13 has been generalized to include row-oriented crowns. Second is that Nilson (1999) assumed a spatially constant within-crown path
length in calculating $P_1$, whereas $P_\ell$ accounts for variable beam path lengths through crowns (Eq. 10).

## A2    Model of Campbell and Norman (1998): OM_VAR

Kucharik et al. (1997) originally suggested a simple empirical model describing the $\theta$-dependence of $\Omega$, provided that the value of $\Omega$ at $\theta = 0$ is known:

$$\Omega(\theta) = \frac{\Omega(0)}{\Omega(0) + (1 - \Omega(0)\exp(-k\theta^p))},$$  (A4)

where $\Omega(0)$ is the value of $\Omega$ when the solar zenith angle is zero, and $k$ and $p$ are geometric coefficients with $p$ given by (Campbell and Norman, 1998)

$$p = 3.8 - 0.46D, \;\; 1 \le p \le 3.34$$  (A5)

and $D$ is the ratio of the crown depth to the crown diameter. The coefficient $k$ is taken to be equal to 2.2 as suggested by Campbell and Norman (1998).
An obvious limitation of this approach is that the value of $\Omega(0)$ must be estimated, which usually requires midday solar interception data. It does, however, make modeling easier since this approach guarantees that the correct radiation interception will be predicted around midday.





## A3   Model of Ni-Meister et al. (2010): NI10_P

Ni-Meister et al. (2010) suggested a geometric model based on spherical crowns, which is an analytical form of the model

originally proposed by Li and Strahler (1988). Their approach used geometry of spheres to back-calculate the appropriate clumping factor $\Omega$, which is given by

$$\Omega = \frac{3}{4\tau_0 r} \left( \frac{1 - (1 - (2\tau_0 r + 1)\exp(-2\tau_0 r))}{2(\tau_0 r)^2} \right), \tag{A6}$$

where

$$\tau_0 r = \frac{3G(\theta)L}{4\lambda \pi R^2}, \tag{A7}$$

and recalling that $G$ is the fraction of leaf area projected in the direction of radiation propagation, $L$ is the canopy leaf area index, $\lambda$ is the number of plants per unit ground area, and $R$ is the crown radius.

Since Ni-Meister et al. (2010) formulated their model in terms of an $\Omega$ clumping factor, the end equations for the models of Ni-Meister et al. (2010) and Nilson (1999) look quite different. However, mathematically they are nearly the same. The primary difference in the formulation is that Nilson (1999) assumes a constant effective path length through crowns in calculating the

probability of leaf intersection, whereas Ni-Meister et al. (2010) explicitly calculates the weighted average probability based on variable path lengths.



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
