# Peer review of "One-dimensional models of radiation transfer in heterogeneous"

_Geoscientific Model Development, 2019_

## Short Comment (SC1) · 16 May 2020

This is an executive editor comment highlighting the ways in which this manuscript is not currently compliant with GMD policy on code and data availability. The issues here must be addressed before a revised manuscript can be accepted for publication.

The archiving of the input and output on Dryad is excellent and represents best practice. However providing a GitHub link for the code is insufficient as GitHub links are neither precise enough nor persistent enough. Instead, the exact version of Helios used in this work needs to be persistently archived. This could be achieved using GitHub's Zenodo.org integration features, or you could simply upload a tarball of the

precise release to Dryad.

Further details on code and data availability requirements are in the GMD model code and data policy: https://www.geoscientific-model-development.net/about/code_ and_data_policy.html. The reasons for the policy and more detail are provided in this editorial: https://doi.org/10.5194/gmd-12-2215-2019.

---

## Referee Comment (RC1) · Anonymous Referee #1 · 21 Jun 2020

**General comments**

This work provides a review and comparison of radiation interception models applied to heterogeneous canopies. The authors compare results from models of varying complexity to data simulated using 3D leaf-resolving model. Though limited to having no scattering, and no diffuse radiation, through the comparison, the authors provide better understand of the effects of model assumptions and inputs. A generalized model is also proposed and shown to perform well compared to other models considered.

**Specific comments**

This work mainly focuses on between crown heterogeneity. Cases simulated mainly

explores densely clumped vegetation with open canopies. Cases 1 and 2 especially focuses on isolating the effects of crown scale clumping. However, heterogeneity with in the crown warrants more discussion. While case 5 using realistic canopy structure includes sub-crown scale clumping, its effects seems overshadowed by the significant crown scale clumping. For example, how will the models behave for a not very dense forest (LAI<2.5) but with mostly closed canopy?

The Leaf angle distribution function (G) is also an important canopy structural characteristic. All cases presented (except 6 perhaps?) use G=0.5. How will other G functions affect results?

Technical corrections

Line 76: should be I(r;s') to be consistent

At Line 94, ah=L. While L is listed in Table 1, please include the physical representation of L (LAI) here for clarification.

Figure 8 caption: Error in legend label for Ni10_P. Document shows 'Ni10_P ((eLine),'.

---

## Referee Comment (RC2) · Anonymous Referee #2 · 26 Jun 2020

This paper contains a well-written discussion of the widely used but poorly defined concept of clumping employed in modelling radiative transfer in plant canopies. The concept is introduced and possible probabilistic approaches to representing it are discussed. These approaches are then assessed using simulations where the reference is a full leaf-resolving model. The binomial model is clearly seen to be the most accurate of those considered. I would be happy to see this paper published and have only a few essentially technical comments.

1. L67. It should say "was explored."

2. L75. The distinction between $s$ and $s'$ seems back to front here. If $s'$ is the incident direction, then $s$ should appear in the first three terms of the equation. Alternatively, if it is the scattered direction, then $s$ and not $s'$ should appear in the radiance in the final term.

3. L111. This should be "incurs".

4. L149, L162. "Augment" suggests an increase. Typically, $\Omega < 1$.

5. Figure 2. The third tree appears to be shown in plan view. This is not explained.
* * *

---

## Author Comment (AC1) · 14 Aug 2020

We graciously thank the reviewers for their critical comments, and careful reading of the manuscript to identify several important typographical errors. We have addressed all comments provided by the reviewers below, and have incorporated them into the revised manuscript.

————————————— Comments by Referee #1 —————————————-

1. This work mainly focuses on between crown heterogeneity. Cases simulated mainly explores densely clumped vegetation with open canopies. Cases 1 and 2 especially

focuses on isolating the effects of crown scale clumping. However, heterogeneity within the crown warrants more discussion. While case 5 using realistic canopy structure includes sub-crown scale clumping, its effects seems overshadowed by the significant crown scale clumping. For example, how will the models behave for a not very dense forest (LAI<2.5) but with mostly closed canopy?

**Reply** We believe that Case 6 illustrates a similar point to what the reviewer is mentioning, but we may not have been sufficiently clear in emphasizing this. Although the geometry for Case 6 is not a tree, it still has significant and realistic sub-crown heterogeneity, as the plant was generated procedurally by following a hierarchical branching structure. The previous version of the manuscript did not directly report the LAI for all geometries in Case 6, so it was probably unclear that the overall LAI for Case 6 was fairly low even when the canopy was mostly closed (L = 0.67). We have added explicit mention of this to the manuscript (lines 336-337).

2. The Leaf angle distribution function (G) is also an important canopy structural characteristic. All cases presented (except 6 perhaps?) use G=0.5. How will other G functions affect results?

**Reply** Case 6 had a highly anisotropic leaf angle distribution, although we acknowledge that the range of values was not explicitly mentioned in the manuscript. Explicit mention of the range of G values for Case 6 was added for clarity and to emphasize the anisotropy for this case (lines 333-334).

While one case of a highly anisotropic G was included, we did not exhaustively explore the effect of G in detail in order to keep the work focused on the effects of heterogeneity, and because our previous work has looked at this in more detail and found that, provided G is symmetric in the azimuth, the impact of G is usually small compared to the impact of heterogeneity (see Ponce de Leon and Bailey 2019 as referenced in the paper). In that work, G was artificially set to be equal to 0.5 even when the actual leaf angle distribution was anisotropic, which had a surprisingly small impact on daily integrated radiation interception. In the present paper, we are substituting the exact value of G into the model, so any impacts of G should be even lower.

As requested by the reviewer, we have added some additional discussion regarding the expected role of G (lines 405-407).

3. Line 76: should be I(r;s') to be consistent

\*\*Reply\*\* Corrected to be semicolon instead of comma.

4. At Line 94, ah=L. While L is listed in Table 1, please include the physical representation of L (LAI) here for clarification.

\*\*Reply\*\* Clarification has been added as suggested: ". . .noting that the leaf area index (LAI) is defined as..".

5. Figure 8 caption: Error in legend label for Ni10_P. Document shows 'Ni10_P ((eLine),'.

\*\*Repy\*\* Corrected.

————————————– Comments by Referee #2 ————————————-

1. L67. It should say "was explored."

\*\*Reply\*\* Corrected.

2. L75. The distinction between s and s′ seems back to front here. If s′ is the incident direction, then s should appear in the first three terms of the equation. Alternatively, if it is the scattered direction, then s and not s′ should appear in the radiance in the final term.

\*\*Reply\*\* The reviewer is correct. The intensity in the final (scattering) term should not have a 'prime' on the s. This has been corrected in the manuscript.

3. L111. This should be "incurs".

**Reply** Corrected.

4. L149, L162. "Augment" suggests an increase. Typically, $\Omega < 1$.

**Reply** We did not mean to imply an increase. This has been re-worded to say: "...and the canopy begins to appear homogeneous with attenuation determined by the value of G$\Omega$L."

5. Figure 2. The third tree appears to be shown in plan view. This is not explained.

**Reply** Clarification has been added to the caption to indicate that two of the trees are shown from the side view, and the other from a top view.

——————————————- Comments by Editor David Ham ——————————————-

We have added source code for Helios v1.0.14 to the permanently archived repository. The code availability statement in the manuscript now reads:

"Helios code version 1.0.14 along with associated project files and output files can be downloaded from the archived repository https://doi.org/10.25338/B85C97. The current version of Helios can be downloaded from https://www.github.com/PlantSimulationLab/Helios."

---

## Author Response (AR1)

Response to Review Comments

We graciously thank the reviewers for their critical comments, and careful reading of the manuscript to identify several important typographical errors. We have addressed all comments provided by the reviewers below, and have incorporated them into the revised manuscript.

**Comments by Anonymous Referee #1**

General comments

This work provides a review and comparison of radiation interception models applied to heterogeneous canopies. The authors compare results from models of varying complexity to data simulated using 3D leaf-resolving model. Though limited to having no scattering, and no diffuse radiation, through the comparison, the authors provide better understand of the effects of model assumptions and inputs. A generalized model is also proposed and shown to perform well compared to other models considered.

Specific comments

This work mainly focuses on between crown heterogeneity. Cases simulated mainly explores densely clumped vegetation with open canopies. Cases 1 and 2 especially focuses on isolating the effects of crown scale clumping. However, heterogeneity with in the crown warrants more discussion. While case 5 using realistic canopy structure includes sub-crown scale clumping, its effects seems overshadowed by the significant crown scale clumping. For example, how will the models behave for a not very dense forest (LAI<2.5) but with mostly closed canopy?

We believe that Case 6 illustrates a similar point to what the reviewer is mentioning. Although the geometry for Case 6 is not a tree, it still has significant and realistic sub-crown heterogeneity, as the plant was generated procedurally by following a hierarchical branching structure. The previous version of the manuscript did not directly report the LAI for all geometries in Case 6, so it was probably unclear that the overall LAI for Case 6 was fairly low even when the canopy was mostly closed ($L = 0.67$). We have added explicit mention of this to the manuscript (lines 336-337).

The Leaf angle distribution function (G) is also an important canopy structural characteristic. All cases presented (except 6 perhaps?) use G=0.5. How will other G functions affect results?

Case 6 had a highly anisotropic leaf angle distribution, although we acknowledge that the range of values was not explicitly mentioned in the manuscript. Explicit mention of the range of $G$ values for Case 6 was added for clarity and to emphasize the anisotropy for this case (lines 333-334).

While one case of a highly anisotropic $G$ was included, we did not exhaustively explore the effect of $G$ in detail in order to keep the work focused on the effects of heterogeneity, and because our previous work has looked at this in more detail and found that, provided $G$ is symmetric in the azimuth, the impact of G is usually small compared to the impact of heterogeneity (see Ponce de Leon and Bailey 2019 as referenced in the paper). In that work, G was artificially set to be equal to 0.5 even when the actual leaf angle distribution was anisotropic,

which had a surprisingly small impact on daily integrated radiation interception. In the present paper, we are substituting the exact value of *G* into the model, so any impacts of *G* should be even lower.

As requested by the reviewer, we have added some additional discussion regarding the expected role of G (lines 405-407).

Technical corrections
Line 76: should be I(r;s') to be consistent

Corrected to be semicolon instead of comma.

At Line 94, ah=L. While L is listed in Table 1, please include the physical representation of L (LAI) here for clarification.

Clarification has been added as suggested: "…noting that the leaf area index (LAI) is defined as..".

Figure 8 caption: Error in legend label for Ni10_P. Document shows 'Ni10_P ((eLine),'.

Corrected.

**Comments by Anonymous Referee #2**

This paper contains a well-written discussion of the widely used but poorly defined concept of clumping employed in modelling radiative transfer in plant canopies. The concept is introduced and possible probabilistic approaches to representing it are dis- cussed. These approaches are then assessed using simulations where the reference is a full leaf-resolving model. The binomial model is clearly seen to be the most accurate of those considered. I would be happy to see this paper published and have only a few essentially technical comments.

1.      L67. It should say "was explored."

Corrected.

2.      L75. The distinction between s and s' seems back to front here. If s' is the incident direction, then s should appear in the first three terms of the equation. Alternatively, if it is the scattered direction, then s and not s' should appear in the radiance in the final term.

The reviewer is correct. The intensity in the final (scattering) term should not have a 'prime' on the s. This has been corrected in the manuscript.

3.      L111. This should be "incurs".

Corrected.

4.     L149, L162. "Augment" suggests an increase. Typically, $\Omega < 1$.

We did not mean to imply an increase. This has been re-worded to say: "…and the canopy begins to appear homogeneous with attenuation determined by the value of $G\Omega L$."

5.     Figure 2. The third tree appears to be shown in plan view. This is not explained.

Clarification has been added to the caption to indicate that two of the trees are shown from the side view, and the other from a top view.

**Comments by Editor David Ham**

The archiving of the input and output on Dryad is excellent and represents best practice. However providing a GitHub link for the code is insufficient as GitHub links are neither precise enough nor persistent enough. Instead, the exact version of Helios used in this work needs to be persistently archived. This could be achieved using GitHub's Zenodo.org integration features, or you could simply upload a tarball of the precise release to Dryad.

We have added source code for Helios v1.0.14 to the permanently archived repository. The code availability statement in the manuscript now reads:

"Helios code version 1.0.14 along with associated project files and output files can be downloaded from the archived repository https://doi.org/10.25338/B85C97. The current version of Helios can be downloaded from https://www.github.com/PlantSimulationLab/Helios."

Response to Review Comments

**List of Relevant Changes Prompted by Reviewer Comments**

1. Changed "were" to "was" on line 67.
2. Added "prime" to s in scattering term of Equation 1.
3. Changed comma to semicolon in I(r;s') on line 76.
4. Bolded 'r' on line 76.
5. Added physical definition of LAI on line 94.
6. Changed "incur" to "incurs" on line 111.
7. Changed "augmented" to "attenuation determined by the value of…" on line 148.
8. Changed "invalidates" to "invalidate" on line 163.
9. Figure 2 caption – added clarification "(two side view, one top view)".
10. Added details regarding range of G values for Case #6 on lines 333-335.
11. Added details regarding LAI values for Case #6 on lines 337-338.
12. Error corrected in Figure 8 caption regarding Ni10_P legend.
13. Brief discussion added regarding anisotropy in G on lines 406-408.
14. "Code and data availability" section updated to reflect availability of source code in repository.

[revised manuscript text omitted]